# SecFPP: Secure Federated Prompt Personalization for Vision Language Models

## Abstract

Prompt learning has emerged as an effective and widely-adopted approach for customizing pre-trained vision language models (VLMs) to user-specific downstream tasks. To tackle data shortage and heterogeneity across multiple users, federated prompt personalization (FPP) has received significant attention as an effective method to harmonize customized performance and pre-trained model generalization capability. However, user-specific prompts, as valuable intellectual assets, face increasing privacy risks such as prompt stealing attacks. Though conventional privacy-preserving techniques such as differential privacy can mitigate these risks by adding noise masks to prompt parameters, they can incur severe performance degradation due to prompt sensitivity. In this work, we propose SecFPP, a secure federated prompt personalization protocol, that reconciles the trade-off among model generalization, local personalization, and privacy preservation. SecFPP delivers state-of-the-art performance under severe data heterogeneity, while using secure multiparty computation primitives to provide formal privacy guarantees without utility loss. The proposed protocol employs a decoupled prompt adaptation strategy by decomposing user prompts into federated and local components, thereby improving personalization performance in multi-granular unbalanced data distributions. We develop a privacy-preserving adaptive clustering algorithm for federated prompts to capture different domains or dataset heterogeneity while using the local prompts to adapt downstream tasks and capture the class heterogeneity. We validate the security of SecFPP theoretically and empirically. Extensive experiments comparing SecFPP with non-private and privacy-preserving baselines demonstrate its superior personalization accuracy. Moreover, comparisons with existing privacy-preserving frameworks highlight that SecFPP significantly improves the privacy-performance trade-off in FPP, simultaneously delivering strongest privacy guarantees and enhanced personalization.

## 1 Introduction

Multimodal large language models (MLLMs) have received significant attention in recent years due to their remarkable generalization capabilities and strong performance in downstream tasks across diverse applications. However, the effectiveness of pre-trained MLLMs in specific downstream tasks is often limited by task-specific data distributions and the need for localized optimization. Prompt tuning has emerged as one of the most effective techniques for improving the performance of pre-trained models Jia et al. (2022); Lester et al. (2021); Liu et al. (2021); Shu et al. (2022). Subsequent works Zhou et al. (2022b;a); Chen et al. (2023); Lu et al. (2022) extended prompt tuning to vision-language models (VLMs) by introducing learnable prompt mechanisms, allowing for more flexible and data-efficient adaptation to downstream tasks. As a lightweight adaptation strategy, prompt learning enables efficient and effective customization of pre-trained models for user-specific downstream tasks. To mitigate local data overfitting and preserve data privacy, researchers have integrated prompt learning into federated learning (FL) frameworks Guo et al. (2023b); Zhao et al. (2023). This integration allows users to collaboratively train prompt parameters while benefiting from the global data distribution. Later works progressively developed the concept of *federated prompt personalization (FPP)*, which has emerged as a promising approach for adapting pre-trained models to individual user tasks through various personalization techniques Guo et al. (2023a); Li et al. (2024); Cui et al. (2024); Yang et al. (2023a); Li et al. (2023).

As in conventional FL, merely keeping data on local devices does not guarantee privacy in FPP. Numerous studies have demonstrated that gradient information leakage in FL can lead to successful privacy attacks (e.g., Geiping et al. (2020); Zhu et al. (2019); Zhao et al. (2020); Li et al. (2022b); Yang et al. (2023b); Petrov et al. (2024); Feng et al. (2024); Du et al. (2024); Zhang et al. (2024); Vu et al. (2024); Das et al. (2025)), and these threats are equally applicable to FPP. To make matters worse, learned prompts usually represent high-value assets, as they include both task-specific knowledge and potentially sensitive user information Shen et al. (2024); Wu et al. (2024); Edemacu & Wu (2024). As in FL (e.g., Dwork (2006); Abadi et al. (2016); Bonawitz et al. (2017); Wei et al. (2020); Shi et al. (2022); Du et al. (2023); Xiao et al. (2025); Demelius et al. (2025)), the most prevalent countermeasure against such attacks in FPP is the application of differential privacy (DP). Consequently, recent FPP studies have also adopted DP as a standard privacy-preserving solution Guo et al. (2023a); Tran et al. (2025). Unfortunately, unlike full machine learning models, the lightweight nature of prompts makes them especially vulnerable to even minor perturbations introduced by DP noise. Under stringent privacy constraints, this sensitivity often results in significant performance degradation. As empirically demonstrated by Tran et al. (2025), DP noise can lead to performance drops of up to 25% under a strict privacy budget, even under low-rank adaption techniques designed to mitigate this effect.

To address the fundamental trade-off between prompt personalization performance and data privacy guarantees, we propose SecFPP, a novel FPP protocol that achieves strong privacy protection without sacrificing personalization performance. Our protocol leverages a secret-sharing primitive, *Lagrange coded computation (LCC)* Yu et al. (2019), and introduces a privacy-preserving prompt clustering mechanism, SecPC. In SecFPP, each user decomposes their prompt into two components: a local prompt retained on-device, and a federated prompt collaboratively learned across users. This decomposition enables granularity-aware prompt adaptation that effectively handles two levels of data heterogeneity. Specifically, federated prompts capture coarse-grained distribution shifts (e.g., cross-domain heterogeneity) that groups of users may have, while local prompts accommodate fine-grained distribution shifts (e.g., inter-label heterogeneity) unique to individual users.

Our contributions are summarized as follows:

- We propose SecFPP, a privacy-preserving protocol for FPP that achieves user-level adaptation through a decoupled prompt structure built upon two key components: LCC and SecPC. This decoupled prompt strategy utilizes a granularity-aware adaptation scheme that balances generalization and personalization across the user federation. The granularity-aware components of the prompt globally adapt to the domain-level heterogeneity while locally accommodating the class-level heterogeneity.

- We develop SecPC, a novel privacy-preserving adaptive clustering algorithm designed for prompt clustering, leveraged to enable effective domain-level adaptation in SecFPP.

- We provide both theoretical and empirical security analyses of SecFPP, demonstrating that the protocol offers strong privacy guarantees with negligible overheads.

- We conduct extensive experiments under diverse data heterogeneity scenarios. Results show that SecFPP achieves state-of-the-art personalization performance, matching or even surpassing existing non-private methods, while significantly outperforming privacy-preserving baselines.

## 2 RELATED WORKS

### 2.1 FEDERATED PROMPT PERSONALIZATION (FPP)

First introduced for vision-language models by Zhou et al. (2022a;b), prompt learning is a lightweight and effective method to enhance the generalization capabilities of foundation models Jia et al. (2022); Lester et al. (2021); Liu et al. (2021); Shu et al. (2022); Lu et al. (2022); Zhou et al. (2022a;b); Chen et al. (2023). Rather than relying on manually engineered prompts, prompt learning introduces parametric prompts, learnable continuous vectors trained on user data. Compared to computationally intensive full-model fine-tuning, prompt learning offers a more efficient alternative with notable gains in downstream performance as shown in Table 1. However, prompts trained solely on local data are prone to overfitting, especially in scenarios with limited or skewed user data distributions. To mitigate overfitting, users can employ federated collaborative prompt learning Guo et al. (2023b); Zhao et al. (2023), jointly training prompt parameters while retaining data locally. The core challenge in this setting is to balance the foundation model's generalization capacity

Table 1: Evolution of prompt learning. Hard prompts refer to manually engineered prompts, such as 'a photo of <class>'. Soft prompts, introduced by CoOp Zhou et al. (2022b;a), are learnable embeddings that were later integrated with the FL framework by PromptFL Guo et al. (2023b) and Fed-Prompt Zhao et al. (2023). Federated prompt personalization (FPP) Guo et al. (2023a); Cui et al. (2024); Li et al. (2024) incorporates personalization techniques to adapt prompts and handle data heterogeneity. DP-FPL Tran et al. (2025) further enhances the privacy of FPP by introducing differential privacy. (Numbers represent accuracy percentages.)

|  | Single Dataset | Multi-Domain Dataset | Privacy Preservation |
| --- | --- | --- | --- |
| Hard Prompt | 68.2 | 61.9 | ✗ |
| Soft Prompt | 91.4 | 84.2 | ✗ |
| FPP | 91.6 | 85.5 | ✗ |
| DP-FPL | 77.4 | 65.7 | ✓ |
| SecFPP (ours) | 91.6 | 91.2 | ✓ |

with user-specific adaptation. Building on the concept of personalization in FL, recent studies have incorporated FL personalization techniques into prompt learning to better navigate this generalization and localization trade-off Zhao et al. (2023); Guo et al. (2023b;a); Li et al. (2023); Yang et al. (2023a); Deng et al. (2024); Li et al. (2024); Cui et al. (2024).

Termed *federated prompt personalization (FPP)*, this approach enables efficient and effective local task adaptation for federated users while maintaining low computational and communication overhead. FedPrompt Zhao et al. (2023) and PromptFL Guo et al. (2023b) are the first to integrate FL into prompt learning through different FL paradigms. Subsequent works, such as pFedPrompt Guo et al. (2023a) and pFedPG Yang et al. (2023a), introduce more practical FPP methods to adapt frozen pre-trained models to local data heterogeneity. pFedPrompt uses a federated prompt and an additional texture encoder for personalized attention to improve local task performance. pFedPG exploits a prompt generator at the server to provide personalized prompts for downstream users. Apart from textual prompts, Li et al. (2023) proposes visual prompts that are attached to image inputs to represent local data distribution. Deng et al. (2024) integrates FL model personalization with prompt selection techniques to resolve data heterogeneity effectively. FedOTP Li et al. (2024) further improves the federated prompt by user consensus knowledge extraction and uses the local prompt for capturing data features in severe data heterogeneity settings, such as label shifts and domain shifts. Addressing a similar challenge, FedPGP Cui et al. (2024) adopts a low-rank prompt decomposition and additional contrastive loss to balance personalization and generalization.

## 2.2 PRIVACY PRESERVATION IN FPP

A personalized prompt is designed to guide a user's downstream task to fully exploit the capabilities of a pre-trained model. As user-specific customization becomes increasingly desirable, personalized prompts for commercially deployed LLMs are now widely recognized as valuable digital assets on various platforms such as OpenAI GPT Store OpenAI, PromptBase PromptBase, Snack-Prompt snackprompt. This growing ecosystem of prompt sharing and reuse introduces significant privacy risks. As high-value assets, customized user prompts for different model architectures have shown vulnerability to various attacks. Notably, prompt stealing attacks Shen et al. (2024) and membership inference attacks Wu et al. (2024) have demonstrated the potential threats of unauthorized prompt usage, reproduction, or even leakage of proprietary or sensitive user data. In FL paradigms, a series of gradient-based privacy attacks have been demonstrated to successfully threaten user prompt information in FPP Geiping et al. (2020); Zhu et al. (2019); Yang et al. (2023b); Petrov et al. (2024); Feng et al. (2024); Du et al. (2024); Zhang et al. (2024); Das et al. (2025).

Although FPP has drawn considerable attention in recent research, its security risks remain rather under-explored, with only a few works addressing privacy concerns. PromptFL Guo et al. (2023b) establishes a connection between differential privacy (DP) in FL and prompt learning. However, it does not effectively adapt DP to the prompt learning task and neglects the need for user-level personalization. A recent work, DP-FPL Tran et al. (2025), is the first to systematically propose a DP-based solution in FPP to seek a potential trade-off among personalization, generalization, and privacy. It implements global and local DP to protect prompts and leverages a low-rank adaption strategy to mitigate performance degradation induced by DP. Following prior work Cui et al. (2024), DP-FPL factorizes prompts into low-rank components to accommodate different data distributions while applying DP noise to low-rank components during training. Nevertheless, empirical results in Table 1 present significant performance degradation (up to $25\%$ loss in certain cases) when a strict

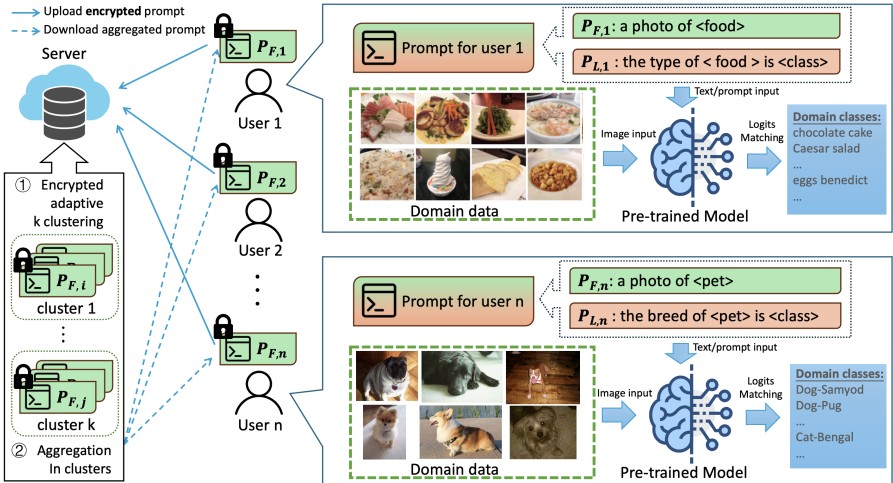

Figure 1: Workflow of SecFPP. On the right, users decompose prompts into federated and local components; the federated prompt adapts to the dataset domain while the local prompt accommodates the local tasks. On the left, the federated prompts are encoded to enable adaptive clustering; then they are aggregated cluster-wise then distributed for the next round.

privacy budget is in place (e.g., $\epsilon \leq 0.01$). Additionally, the personalization design of DP-FPL falls short of adapting to domain-level data heterogeneity, as only the local prompt is factorized for personalization while the federated prompt is learned universally.

## 3 PROBLEM OVERVIEW

In this section, we present the problem formulation, followed by the threat model and overview of the proposed protocol, SecFPP.

### 3.1 PROBLEM FORMULATION

Let $F(\cdot)$ denote an upstream pre-trained model. In FPP, orchestrated by a central server, $n$ end users collaboratively train personalized prompts to adapt the pre-trained model to their datasets while fully utilizing the model's generalization capacity without overfitting. Denote user local datasets by $\{\mathcal{D}_1, \mathcal{D}_2, \ldots, \mathcal{D}_n\}$, where user $i$ has a distinct dataset $\mathcal{D}_i$. Instead of using pre-defined prompts (e.g., "a photo of $\langle label \rangle$"), users train their own prompts as learnable embeddings (or soft prompts), $\mathbf{P}_i \in \mathbb{R}^{d \times k}$, where $d$ is the dimension of the word embedding and $k$ is the number of tokens. We only keep the learnable parts in prompt and omit the hard prompt tokens with masked label positions for simplicity in formulations.

Consider different levels of data heterogeneity caused by data domains or an unbalanced category split within a single domain. We denote personalized prompt of user $i$ by $\mathbf{P}_i$. It is decomposed into federated prompts and local prompts by $\mathbf{P}_i = [P_{F,i}, P_{L,i}]$. $P_{F,i}$ is trained federatively while $P_{L,i}$ is kept private by each user. In a downstream task, each user $i$ performs classification by $\hat{y} = F(x, \mathbf{P}_i)$, for $\forall (x, y) \in \mathcal{D}_i$. Hence, the optimization objective for each user is given by:

$$\mathcal{L}_{\mathcal{D}_i} = \sum_{(x,y) \in \mathcal{D}_i} \ell_{ce}(y, F(x, \mathbf{P}_i)), \tag{1}$$

where $\ell_{ce}$ stands for the cross entropy loss of the prediction $\hat{y}$ and $y$. The overall optimization problem for user prompts is:

$$\arg\min_{\{\mathbf{P}_i | i \in [n]\}} \sum_{i \in [n]} \frac{1}{|\mathcal{D}_i|} \mathcal{L}_{\mathcal{D}_i}, \tag{2}$$

where $[n]$ denotes the user index set $\{1, \ldots, n\}$ and $|\cdot|$ denotes the set cardinality.

### 3.2 THREAT MODEL

The server and the users are *honest-but-curious*, which is commonly used in evaluating the privacy of FL protocols Geiping et al. (2020); Bonawitz et al. (2017); So et al. (2022); Buyukates et al. (2024); Tran et al. (2025). That is, all parties in the system follow the prescribed protocol faithfully, but the curious server and colluding users attempt to reveal private information of prompts, from both their local states and received messages during the protocol execution. We assume that the

---

**Algorithm 1:** SecFPP: Secure Fed-Prompt Personalization

---

**Data:** Local datasets $\{\mathcal{D}_1, \ldots, \mathcal{D}_n\}$ distributed on $n$ users
**Input:** Communication round $T$, learning rate $\eta$, decomposition rank $k$.
**Output:** Personalized prompts $\mathbf{P}_i$ for users

1 Initialize cluster assignment $\mathcal{S} = \{[n]\}$;

2 Initialize personalized prompt parameters $\mathbf{P}_i^{(0)} = \left[ P_{F,s}^{(0)}, P_{L,i}^{(0)} \right], s \in \mathcal{S}$;

3 **for** *iteration* $t \leftarrow 0$ **to** $T-1$ **do**

4     **for** *user* $i \in [n]$ *in parallel* **do**

5        Compute loss by Eq.1 and calculate gradients as $\nabla_{G,i}\mathcal{L}$ and $\nabla_{L,i}\mathcal{L}$ correspondingly;

6        Update local prompt by $p_{L,i}^{(t+1)} \leftarrow p_{L,i}^{(t)} - \eta \nabla_{L,i}\mathcal{L}$;

7        Use truncated SVD to reduce dimension for the personalized prompt $\overline{\mathbf{P}}_i \leftarrow PCA(\mathbf{P}_i^{(t+1)}, k)$ ;

8     **end**

9     Perform privacy-preserving adaptive $k$-means SecPC on reduced personalized prompts,
      $\mathcal{S} \leftarrow SecPC(\mathcal{S}, \{\overline{\mathbf{P}}_i \mid i \in [n]\})$;

10    Use secure aggregation for federated prompt gradients according to cluster assignment, equivalently
      as, $\nabla_{G,s}\mathcal{L} \leftarrow \frac{1}{|s|} \sum_{i \in s} \nabla_{G,i}\mathcal{L}, \quad \forall s \in \mathcal{S}$;

11    Update clustered federated prompts by $P_{F,s}^{(t+1)} \leftarrow P_{F,s}^{(t)} - \eta \nabla_{G,s}\mathcal{L}, \quad \forall s \in \mathcal{S}$;

12    Users update personalized prompts by $\mathbf{P}_i^{(t+1)} \leftarrow \left[ P_{F,s|i \in s}^{(t+1)}, P_{L,i}^{(t+1)} \right], \quad \forall i \in [n]$.

13 **end**

---

adversary cannot control both the server and a subset of users simultaneously, as is common in the secure federated learning literature Bonawitz et al. (2017); So et al. (2020a); Shao et al. (2022).

### 3.3 SOLUTION OVERVIEW

Prior works in FPP are limited in two aspects: they mostly address a single source of data heterogeneity, and whereas they focus on improving personalization but overlook the privacy protection of user prompts. Existing differentially private FPP works also heavily suffer from model degradation and poor domain adaptation. For the first aspect, we observe that the source of data heterogeneity significantly affects the distribution of user prompts. In scenarios with only class skew, the distribution difference in prompts aligns with the class distribution. In contrast, when both class skew and data from different domains are present, the disparity in prompt distributions is predominantly attributed to domain differences, with the effect of class skew becoming relatively minor. Hence, to address the critical challenges in performance degradation and privacy preservation in FPP, we propose a secure federated prompt personalization protocol, SecFPP, that integrates a privacy-preserving adaptive clustering algorithm and decoupled prompt personalization scheme.

In the ensuing sections, we first introduce the cryptographic building blocks including the coding primitive, LCC, and the privacy-preserving adaptive clustering algorithm, SecPC; then present the SecFPP workflow in detail; and at last, provide the theoretical analysis for the security of the protocol. The overview of SecFPP is presented in Figure 1.

## 4 PROPOSED SECURE FPP: SECFPP

### 4.1 CRYPTOGRAPHIC BUILDING BLOCKS

SecFPP involves an innovative privacy-preserving clustering algorithm, SecPC. Both clustering and aggregation are built upon the coding primitive LCC.

**SecPC.** We propose a secure adaptive prompt clustering algorithm presented in Algorithm 2, which is developed upon the well-known $k$-means clustering algorithm and its adaptive variants MacQueen (1967); Darken & Moody (1990); Bhatia et al. (2004); Xia et al. (2020). Since the prompt clustering is agnostic to the overall data distribution, the number of clusters is unknown initially. We build upon a simple yet effective adaptive $k$-means algorithm Bhatia et al. (2004) to perform clustering on user prompts federatively. During this clustering, only the relative distances to the cluster centers are communicated and revealed to the server, for which we present the security analysis in Section 4.3. Different from Bhatia et al. (2004), we do not update the centers whenever a data point is assigned. Instead, SecPC only performs cluster merging and center updating once at the end of each communication round. In this way, the clustering iterations correspond to the FL communication rounds (which can be considered as a one-shot clustering).

---

**Algorithm 2:** SecPC: Secure Prompt Clustering

---

**Data:** Reduced local prompts $\overline{\mathbf{P}}_i$ of $n$ users, previous round cluster assignment $\mathcal{S}$.

**Output:** Updated $k$-cluster assignment $\mathcal{S} : \{s_1, \ldots, s_k\}$.

1   Server broadcasts previous clustering $\mathcal{S}$ ;

2   **for** *user $i \in [n]$ in parallel* **do**

3      Slice its reduced local prompt $\overline{\mathbf{P}}_i$ by length $\ell$ and generate secret shares by LCC sharing scheme,
$\left\{ \left[ \overline{\mathbf{P}}_i \right]_1, \ldots, \left[ \overline{\mathbf{P}}_i \right]_n \right\} \leftarrow LCC.Share(\overline{\mathbf{P}}_i)$;

4      Share $\left[ \overline{\mathbf{P}}_i \right]_j$ with user $j \in [n]$;

5   **end**

6   **for** *user $j \in [n]$ in parallel* **do**

7      Update local coded center, $[\mu_{\mathbf{s}}]_j \leftarrow \sum_{i \in s} \left[ \overline{\mathbf{P}}_i \right]_j, \forall s \in \mathcal{S}$ ;

8      Compute coded distances for all users to each center
$[d_{i,s}]_j \leftarrow \left\| [\mu_{\mathbf{s}}]_j - |s| \cdot \left[ \overline{\mathbf{P}}_i \right]_j \right\|_2^2, \quad \forall (i, s) \in [n] \times \mathcal{S};$

9      Transmit all coded distances to the server;

10   **end**

11   Server reconstructs each $d_{i,s}$ by Reed-Solomon decoding,
$d_{i,s} \leftarrow LCC.recon(\{[d_{i,s}]_j \mid j \in [n]\}), \forall (i, s) \in [n] \times \mathcal{S}$ ;

12   Server recovers the real distances by $d_{i,s} \leftarrow \frac{d_{i,s}}{|s|^2}, \forall (i, s) \in [n] \times \mathcal{S}$;

13   Server performs one-shot adaptive clustering and updates cluster assignment $\mathcal{S}$.

---

**LCC**. The key preliminary to achieve secure clustering is the secret-sharing scheme named Lagrange coded computing (LCC) Yu et al. (2019). LCC is a multi-secret sharing primitive using Lagrange polynomial interpolation. The decryption of LCC is robust to missing and erroneous computation results with Reed-Solomon decoding. Compared with Shamir secret sharing Shamir (1979), LCC reduces the share size and, thus, the load of computation on each party by $\ell$ times. We denote the usages of the sharing and reconstruction algorithms by $LCC.share(\cdot)$, $LCC.recon(\cdot)$, respectively.

**Secure Aggregation**. SecFPP also involves an aggregation step that group-wise aggregates federated prompt components according to the cluster assignments $\mathcal{S}$. We employ the existing off-the-shelf secure aggregation technique based on the same primitive of LCC, such as So et al. (2022); Jahani-Nezhad et al. (2023); Buyukates et al. (2024); Hou et al. (2024).

### 4.2 SECFPP WORKFLOW

In the proposed SecFPP scheme, we introduce an effective federated prompt personalization protocol that provides privacy guarantees while addressing multi-level data heterogeneity, presented in Figure 1. In prior FPP works, various approaches are proposed to balance generality and local adaptation Li et al. (2024); Cui et al. (2024); Tran et al. (2025), but these methods all adopt a split structure of personalized prompt comprising a global prompt (shared across clients) and a local prompt (customized for individual users). In practice, data heterogeneity may come from diverse sources. It may arise from pathological class distributions within a single dataset but also may fundamentally come from distinct types of datasets. Motivated by such split structure, we propose to use the splitting decomposition to achieve prompt adaptation for multi-source data heterogeneity: a federated prompt component is dynamically adjusted to domain heterogeneity, and a local prompt component adapts local heterogeneity while accommodating different user downstream tasks. To provide privacy preservation to domain-level personalization, we develop SecPC to achieve a simple yet effective approach to adaptively cluster the federated prompt component.

At the beginning of training, the server initializes a universal prompt with global and local components for all users. In each communication round, each user computes the gradients for each component using loss $\mathcal{L}_{\mathcal{D}_i}$ measured on the local dataset $\mathcal{D}_i$. After finishing local training epochs, each user updates its local prompt. After updating the personalized prompt $\mathbf{P}_i$ by the local prompt, it performs dimensional reduction on the personalized prompt to result in a prompt summary, using the truncated SVD or PCA Wold et al. (1987). The dimensional reduction preserves the most important components of the prompt while greatly reducing the transmitted data. The adaptive clustering, SecPC, is then performed federatively on the reduced $\overline{\mathbf{P}}_i$ and results in a cluster assignment, denoted by $\mathcal{S}$. According to the cluster assignment, the server invokes a round of secure aggregation to cluster-wise aggregate the gradients for global components and back-propagates for the federated prompts. Finally, the federated prompt component is updated into the personalized prompt for the next round. We summarize the protocol of SecFPP in Algorithm 1.

### 4.3 THEORETICAL ANALYSIS

As shown in Algorithms 1 and 2, $P_{L,i}$ is always privately kept by users while $\overline{\mathbf{P}}_i, P_{F,s}$ join the secret-sharing based algorithms. LCC preserves perfect secrecy for the coded information within the security threshold of shareholders. Hence, we are interested in the theoretical analysis for the revealed information in the protocol, specifically, the reconstructed distances $d_{i,s}$, cluster assignment $\mathcal{S}$ and federated prompts. Cluster assignment is non-parametric and federated prompts are considered public, hence, we focus on analyzing reconstructed distances on the curious server.

We assume that the prompt vector of each user has independent and identically distributed (i.i.d.) entries and satisfies a Gaussian distribution of $\mathcal{N}(\mu_i, \sigma_i)$. Despite its simplicity, this assumption stands because PCA-reduced prompts exhibit a robust approximately sub-Gaussian distribution in practice. We denote prompt's vector space by $\mathcal{P}$. Without loss of generality, we analyze a single user prompt $\overline{\mathbf{P}}_i$ with one cluster of prompts, $\{\overline{\mathbf{P}}_1, \ldots, \overline{\mathbf{P}}_n\}$, such that $\overline{\mathbf{P}}_j \sim \mathcal{N}^d(\mu_j, \sigma_j), \forall j \in [n]$, where $i$ may or may not be in $[n]$. We denote the average over the cluster by $\overline{\mathbf{P}}_{avg}$, which is the cluster center. To quantify the information leakage by the distance to the center, we study the mutual information between a prompt and the distance in the following theorem.

**Theorem 1.** *Given a cluster of prompts as normal random vectors by $\overline{\mathbf{P}}_j \sim \mathcal{N}^d(\mu_j, \sigma_j), j \in [n]$, the distance is the $\ell_2$-norm between a prompt $\overline{\mathbf{P}}_i$ and the cluster center $\overline{\mathbf{P}}_{avg}$, i.e., $D^2 = \left\| \overline{\mathbf{P}}_i - \overline{\mathbf{P}}_{avg} \right\|_2^2$. The mutual information between $\overline{\mathbf{P}}_i$ and $D^2$ is given by:*

$$
\begin{aligned}
MI\left(\overline{\mathbf{P}}_i; D^2\right) = {}& \log 2\Gamma\left(\frac{d}{2}\right) + \left(1 - \frac{d}{2}\right)\psi\left(\frac{d}{2}\right) + \frac{d}{2} \\
& + \int \cdots \int_{\mathbf{P} \in \mathcal{P}} f_{\overline{\mathbf{P}}_i}\left(\overline{\mathbf{P}}_i = \mathbf{p}\right) \cdot \begin{cases} \left(\ln(2) + h_d\left(\frac{\tau}{2}\right) + c\right) \cdot d\mathbf{P} & \text{if } d \in \mathbb{N}^{odd}, \\ \left(\ln(2) + g_{d/2}\left(\frac{\tau}{2}\right) + c\right) \cdot d\mathbf{P} & \text{if } d \in \mathbb{N}^{even}. \end{cases}
\end{aligned} \tag{3}
$$

*where $\Gamma, \psi$ represent gamma function and digamma function, respectively; $f_{\overline{\mathbf{P}}_i}$ is the probability density function of $\overline{\mathbf{P}}_i$; $c$ is $2\log\left(\frac{n-1}{n}\right)$; $h_m$ and $g_n$ are families of functions expanded in appendix.*

*Proof.* See Appendix A.3. $\qquad\square$

**Remark 1**: For rare edge cases, there is an infinitesimal possibility that one could reconstruct a user's entire prompt solely from distance measurements. However, when the prompts are randomly distributed, such reconstruction becomes statistically infeasible. Our focus is on typical scenarios and we aim to statistically answer the following question: *To what extent can an honest-but-curious server infer information about a user's prompt from reconstructed distances?* In Theorem 1, we provide an analytical formulation that characterizes the mutual information between a user's reduced prompt representation $\overline{\mathbf{P}}_i$ and the squared distance $D^2$, which quantifies the information leakage from the distance in a rigorous, statistical sense. Although deriving a tight upper bound on this mutual information is intractable, the expression serves as a theoretical foundation for analyzing privacy leakage. For instance, if $MI\left(\overline{\mathbf{P}}_i; D^2\right) << h\left(\overline{\mathbf{P}}_i\right)$, the distance provides negligible information about the prompt, i.e., observing $D^2$ reduces only an infinitesimal amount of uncertainty in the prompt. In such cases, the system satisfies information-theoretic privacy guarantees. To demonstrate that, we present empirical mutual information estimations in Section 5.3.

**Remark 2**: Equation (3) has two main variables: $d$ and $n$. $MI(\overline{\mathbf{P}}_i; D^2)$ is dominated by the prompt dimension $d$ as the degrees of freedom. Though $n$ is a variable in the integral of constant, $\log\frac{n-1}{n}$ approaches zero when $n$ is large. The function family $h_n$ and $g_m$ exhibits logarithmic characteristics with strictly increasing monotonicity and is increasingly monotonic in $d$. In Section 5.3, we numerically demonstrate these characteristics.

## 5 EXPERIMENTS

In this section, we evaluate the FPP performance of SecFPP and compare it with private and non-private baselines under different levels of data heterogeneity. Moreover, we present empirical results for the mutual information estimations in the security analysis in Section 4.3. Finally, we perform an evaluation of the computational overheads to demonstrate the cost of privacy in SecFPP.

### 5.1 EXPERIMENTAL SETTINGS

Following previous works Tran et al. (2025); Cui et al. (2024); Li et al. (2024); Guo et al. (2023a), we perform the prompt-based image classification tasks on the pre-trained CLIP model Radford

Table 2: FPP performance with non-private and privacy-preserving protocols (accuracy in %).

| Personalization Methods | Single Datasets | | | | | | Multi-Domain Datasets | | | | | | | |
|---|---|---|---|---|---|---|---|---|---|---|---|---|---|---|
| | CIFAR-10 | Caltech-101 | Oxford-Pet | Oxford-Flowers | Food-101 | DTD | CIFAR-10+Caltech-101 | CIFAR-100+Caltech-101 | Caltech-101+Oxford-Pet | Caltech-101+Oxford-Flowers | Caltech-101+Food-101 | Caltech-101+DTD | Oxford-Pet+Oxford-Flowers | Food-101+DTD |
| PromptFL | 89.0 | 91.4 | 82.8 | 69.8 | 70.6 | 42.4 | 89.1 | 74.8 | 84.1 | 78.3 | 74.9 | 65.1 | 77.2 | 54.1 |
| FedOTP | **89.6** | **91.6** | **86.7** | 66.4 | **74.8** | **44.6** | **89.8** | 74.7 | 85.4 | **79.4** | 84.1 | **68.9** | 78.1 | **59.3** |
| FedGPG | 88.9 | 91.1 | **86.8** | **70.1** | 72.2 | 42.6 | 88.6 | 74.0 | **91.4** | 76.3 | 75.5 | 67.8 | **81.2** | 57.8 |
| DP-FPL w.o. privacy | 89.3 | 90.9 | 84.9 | **70.2** | 71.7 | 41.9 | 89.2 | 76.0 | 85.3 | 77.6 | 80.6 | 67.1 | 80.8 | 56.7 |
| DP-FPL w. loose privacy | 88.5 | 85.6 | 82.2 | 66.2 | 68.9 | 35.5 | 83.4 | 70.6 | 78.5 | 74.5 | 72.5 | 63.0 | 75.5 | 53.1 |
| DP-FPL w. default privacy | 86.1 | 77.4 | 77.6 | 61.2 | 66.3 | 32.6 | 82.9 | 62.2 | 73.1 | 53.1 | 66.3 | 45.4 | 64.0 | 49.1 |
| DP-FPL w. strict privacy | 82.6 | 74.7 | 44.8 | 27.4 | 52.2 | 22.0 | 71.5 | 50.8 | 65.7 | 56.7 | 54.9 | 41.3 | 35.4 | 45.4 |
| SecFPP (ours) | **89.4** | **91.6** | 86.3 | **70.6** | 74.7 | **44.7** | **90.6** | **77.8** | 91.2 | **79.6** | **87.6** | **69.8** | 82.7 | **59.4** |

et al. (2021) using ViT-B/16 as backbone Dosovitskiy et al. (2020). The implementation details can be found in Appendix A.2.

**Datasets**. We consider various datasets from different domains to evaluate the FPP tasks. We use general-domain datasets: CIFAR-10, CIFAR-100 Krizhevsky et al. (2009) and Caltech-101 Li et al. (2022a); along with specific-domain datasets: Oxford-Pet Parkhi et al. (2012), Oxford-Flowers Nilsback & Zisserman (2008), Food-101 Bossard et al. (2014), and a texture database DTD Cimpoi et al. (2014). To comprehensively simulate data heterogeneity, we consider multi-granular heterogeneity for data distribution. For single dataset allocation, we split the dataset evenly across users by Dirichlet distribution. Then, we allocate two domains of datasets to each half of the users. Within each dataset, we also apply Dirichlet distribution as the non-i.i.d. partition. This dual-level heterogeneity is denoted as datasetA + datasetB in the following section, e.g., Caltech101+OxfordPets.

**Baselines**. For non-private baselines, we consider PromptFL Guo et al. (2023b), FedOTP Li et al. (2024), and FedGPG Cui et al. (2024). PromptFL is a federated version of CoOp Zhou et al. (2022a). FedOTP and FedGPG are the existing state-of-the-art FPP protocols. Regarding privacy-preserving schemes, DP-enabled PromptFL is not capable of training personalized prompts, resulting in poor performance. Hence, we consider the only existing privacy-preserving FPP scheme, DP-FPL Tran et al. (2025), for comparisons.

## 5.2 FPP Performance Comparisons

The overall performance is presented in Table 2. In general domain datasets, such as CIFAR-10, Caltech-101, all baselines have marginal differences. In contrast, for single specific-domain datasets, FPP-based methods consistently achieve higher accuracy compared to non-personalized PromptFL. SecFPP also achieves comparably strong accuracy results in all single-domain datasets. In multi-domain datasets, the three non-private FPP approaches demonstrate distinct advantages on different dataset combinations while SecFPP presents consistent advantages in accuracy. When the combined datasets are general (such as CIFAR-10+Caltech-101), all methods demonstrate comparable accuracy levels, although our solution exhibits a marginal but consistent performance advantage. Notably, in scenarios with high domain discrepancy, SecFPP's advantage becomes particularly pronounced. While FedOTP demonstrates competitive robustness under severe data heterogeneity compared to other FPP approaches, SecFPP consistently surpasses all existing FPP solutions in these challenging scenarios.

On the other hand, for the private FPP baselines, while DP-FPL can achieve comparable performance to non-private solutions without DP noise, its accuracy significantly degrades when DP is applied. Furthermore, under tighter privacy constraints, DP-FPL's performance deteriorates proportionally. In sharp contrast, our SecFPP provides rigorous privacy guarantees without compromising any model performance, while demonstrating superior robustness in personalization for both single-domain and multi-domain heterogeneous data scenarios.

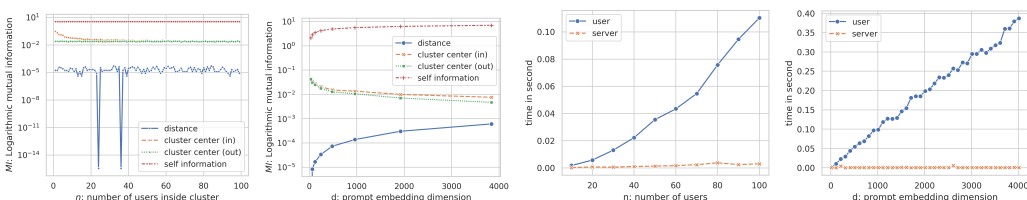

(a) MI vs. user number     (b) MI vs. dimension     (c) time vs. user number     (d) time vs. dimension

Figure 2: (a)(b) are mutual information comparisons. Legend entries are $MI(\overline{\mathbf{P}}_i; D^2)$, $MI(\overline{\mathbf{P}}_i; \overline{\mathbf{P}}_{avg})$ with prompt inside cluster, $MI(\overline{\mathbf{P}}_i; \overline{\mathbf{P}}_{avg})$ with prompt outside cluster, self-information $h(\overline{\mathbf{P}}_i)$ respectively. (c)(d) are computational cost in seconds per round.

### 5.3 Empirical Studies for Security

In this subsection, we present numerical analysis for mutual information (MI) following the analytical results in Theorem 1. We exploit widely used KSG algorithm Kraskov et al. (2004) for estimations and refer to previous studies with MI estimation Ross (2014); Gao et al. (2018); Wang et al. (2021). Recall in Section 4.3, we aim to answer '*how much information about the prompt an honest-but-curious server can infer from the reconstructed distances?*'. Hence, the distance's statistical insignificance with respect to any reduced prompt is presented. We consider three variables for estimating MI as comparisons: entropy of a user prompt (self MI), $h(\overline{\mathbf{P}}_i)$; MI between a user prompt and cluster center, $MI(\overline{\mathbf{P}}_i; \overline{\mathbf{P}}_{avg})$; MI between a user prompt and the distance, $MI(\overline{\mathbf{P}}_i; D^2)$. For $MI(\overline{\mathbf{P}}_i; \overline{\mathbf{P}}_{avg})$, we consider both cases that the prompt is inside or outside the given cluster. As in Gao et al. (2018), numerical analysis shows that a sample size over 1000 reduces mean squared residuals to $10^{-3}$, ensuring stable estimation. So, we use 1000 as sample size and sample user prompts as i.i.d. random vectors. Specifically, each prompt has 15 tokens. The reduced dimension is 8 and original dimension is 512 as default, and $(\mu_i, \sigma_i)$s are simplified to $(0, 1)$. For the distance as a number, we directly use the deterministic scalar-to-vector mapping (i.e. replicating $D^2$ to span dimensions) for MI estimation, such that the entropy is preserved by the deterministic function. The evaluation results are presented in Figure 2 (a), (b).

As demonstrated by the blue trajectory in Figures 2a and 2b, the prompt dimension $d$ is the predominant factor and exhibits a positive correlation with MI, while the user number $n$ has negligible influence (where $n = |\mathcal{S}|$). The empirical results align with the analytical result in Equation (3). In both evaluations, $MI(\overline{\mathbf{P}}_i; D^2)$ between the distance and the prompt (blue) remains exponentially lower than the prompt information entropy (red). Specifically, given $\log_{10} \frac{h(\overline{\mathbf{P}}_i)}{MI(\overline{\mathbf{P}}_i; D^2)} > r$ with $r$ as a lower bound, $r = 4$ is observed such that $MI(\overline{\mathbf{P}}_i; D^2) << h(\overline{\mathbf{P}}_i)$. This substantial gap ($r > 2$) provides rigorous information-theoretic constraint on any adversarial attempt. Thus, *SecFPP provides information-theoretic privacy guarantees for user prompts in practical settings.*

We also present mutual information $MI(\overline{\mathbf{P}}_i; \overline{\mathbf{P}}_{avg})$ between a given prompt and the cluster center for further illustrations. When a prompt resides within a cluster and the cluster cardinality is small, the cluster center exhibits higher statistical dependence on the prompt, inducing larger $MI(\overline{\mathbf{P}}_i; \overline{\mathbf{P}}_{avg})$. The dependency diminishes asymptotically by increasing cluster cardinality $|\mathcal{S}|$ and dimension $d$. Generally, the observation agrees with the expectation that the cluster center vector conveys more information than the scalar distance while diluting the original prompt entropy by averaging over more prompts. Thus, the fact that SecFPP only reveals relative distances, rather than explicit cluster centers, offers inherently stronger security guarantees, as it significantly limits the server's ability to infer sensitive prompt information.

For the additional computation and communication costs of achieving SecFPP with privacy guarantees, Figure 2 (c) and (d) present the empirical overheads of performing SecPC in with dominant factors: $n$ and $d$, which exhibits a linear trend and remains a low practical cost. The detailed analysis on SecFPP complexity and overhead breakdowns are presented in Appendix A.4.

## 6 Conclusion

In this paper, we present a novel secure federated prompt personalization protocol, SecFPP, that addresses data heterogeneity from multiple sources via decoupling prompts and privacy-preserving adaptive clustering, SecPC. Extensive experiments validate SecFPP's robust performance across various data unbalance, demonstrating consistent and superior performance over SOTA baselines. Moreover, SecFPP theoretically guarantees and empirically validates strong privacy preservation on the user prompts, bridging a critical gap in the performance-privacy trade-off of FPP schemes.

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

# A APPENDIX

## A.1 ADDITIONAL PRELIMINARIES

In SecFPP, the operations of data sharing and distance computation are carried out in a finite field $\mathbb{F}_q$, for some large prime $q$. Hence, for a data point $\mathbf{x}_i$ from the real field, one needs to first quantize it onto $\mathbb{F}_q$. We abuse the symbols only in this section.

**Data Quantization.** The quantization technique in SecFPP has been utilized in So et al. (2020a) So et al. (2022). To quantize a data point $\mathbf{x}_i$, we first scale it by $\lambda$, and embed the scaled value onto $\mathbb{F}_q$ such that

$$\bar{\mathbf{x}}_i = \mathcal{Q}(\mathbf{x}_i, \lambda) = \begin{cases} \lfloor \lambda \mathbf{x}_i \rceil, & \text{if } \mathbf{x}_i \geq 0 \\ \lfloor q + \lambda \mathbf{x}_i \rceil, & \text{if } \mathbf{x}_i < 0 \end{cases} . \tag{4}$$

Here $\lfloor x \rceil$ denotes the largest integer less than or equal to $x$, and the quantization function $\mathcal{Q}$ is applied element-wise. Assuming each element of $\lambda \mathbf{x}_i$ is within $[-\eta, \eta)$ for some $\eta > 0$, then on the range of $\mathcal{Q}$, $[0, \eta)$ is the image of the positive part, and $[q - \eta, q)$ is the image of the negative part. While $\lambda$ controls the precision loss of quantization, it also needs to be chosen so that overflow does not occur during computations of SecFPP. A larger $\eta$ requires a larger finite field size $q$ to avoid computation overflow, and a smaller $\eta$ leads to a higher precision loss between $\mathbf{x}_i$ and $\bar{\mathbf{x}}_i$. We can choose a proper scaling factor $\lambda$ to preserve enough precision while keeping the field size practical. To avoid computation overflow, we should choose $q$ such that all the intermediate computation results on the scaled data $\lambda \mathbf{x}_i$ are within the range $\left(-\frac{q}{2}, \frac{q}{2}\right)$. In the worst case, the largest distance across data points and cluster centers $D \triangleq \max_{i \in [m], h \in [k]} \|\boldsymbol{\mu}_h - |\mathcal{S}_h| \cdot \mathbf{x}_i\|_2^2$ results in the largest output value. Therefore, we should choose $q$ that is at least $2\lambda^2 D$.

**Lagrange Coded Computing (LCC).** The technique used when each client secret shares its data with the other clients is Lagrange Coded Computing (LCC). As a result, each client gets a hold of the entire dataset in a coded manner, thereby being able to compute the secret shares of the distances between each pair of datapoints and cluster centers. Similar secret-sharing operations between clients have been widely utilized in FL literature to improve performance, privacy, and robustness (see e.g., Shao et al. (2022), Schlegel et al. (2023), So et al. (2020a), Bonawitz et al. (2017), So et al. (2020b)). In Shao et al. (2022), authors tackle the issue of non-iid data distribution of local client datasets, and use LCC-based secret sharing of client data to improve the convergence of FL models. In Reference Schlegel et al. (2023), authors use Shamir's secret sharing to encode and share their local datasets with the other clients. The added redundancy in the encoding process helps mitigate stragglers in FL, without sacrificing local data privacy. In Reference So et al. (2020a), clients exchange secret shares of their local model updates in a verifiable manner during FL iterations, so that each client can make sure that others have sent valid secret shares, which are then utilized by the server to decode the pairwise distances among client models for outlier detection and removal before aggregation. In Bonawitz et al. (2017), secret shares of the client models are utilized to handle client dropouts during secure model aggregation. Authors of So et al. (2020b) study the setting of decentralized FL, and utilize secret shares of private datasets among multiple parties to perform distributed training, while still preserving privacy.

## A.2 IMPLEMENTATION DETAILS

The default number of users, maximum number of epochs, number of local epochs, and learning rate are $20, 100, 10$, and $0.001$, respectively. For protocols involving low-rank decomposition, we set the default rank to $8$. In SecFPP, we set the first-k principal components to $8$ as well.

For DP-FPL, we choose DP parameter $\epsilon$ from $\{0.0, 0.4, 0.1, 0.01\}$ as no privacy, loose privacy, default privacy, and strict privacy constraints, respectively. We set the Dirichlet distribution parameter to $\beta = 0.3$.

All experiments are conducted on a machine using Intel(R) Xeon(R) Gold 5118 CPU @ 2.30GHz, with 12 cores of 48 threads. The computational capability of a user is constrained to one-quarter of the server by parallelization.

### A.3 SUPPLEMENTARY TO PROOF

*Proof.* (Part I) We are given reduced prompts $\{\overline{\mathbf{P}}_1, \ldots, \overline{\mathbf{P}}_n\}$ as d-dimensional continuous random vectors in a cluster, with sample mean $\overline{\mathbf{P}}_{avg} = \frac{1}{n} \sum_{j=1}^n \overline{\mathbf{P}}_j$. As all prompts satisfy $\overline{\mathbf{P}}_j \sim \mathcal{N}^d(\mu_j, \sigma_j)$, their average is also normally distributed, $\overline{\mathbf{P}}_{avg} \sim \mathcal{N}^d(\mu_{avg}, \sigma_{avg})$, where $\mu_{avg}, \sigma_{avg}^2$ are linear combinations of prompts' $\mu$ and $\sigma^2$. For a prompt $\overline{\mathbf{P}}_i$ inside or outside the given cluster, the distance between the prompt and the cluster center is $D^2 = \left\| \overline{\mathbf{P}}_i - \overline{\mathbf{P}}_{avg} \right\|_2^2$. We discuss the inside case in this part of the proof and the outside case in Appendix A.3 as Part II of proof.

Expand the mutual information in the form of differential entropy:

$$MI\left(\overline{\mathbf{P}}_i; D^2\right) = h\left(D^2\right) - h\left(D^2 \mid \overline{\mathbf{P}}_i\right). \tag{5}$$

For the entropy $h(D^2)$ we make the following observation: $D^2$ is simply a $\chi^2$-distribution with $d$ degrees of freedom, since it is the summation of the squares of all entries with centered normal distribution. Then:

$$h(D^2) = \log 2\Gamma\left(\frac{d}{2}\right) + \left(1 - \frac{d}{2}\right)\psi\left(\frac{d}{2}\right) + \frac{d}{2}, \tag{6}$$

where $\Gamma$ and $\psi$ represent gamma function and digamma function, respectively. For the conditional entropy:

$$h\left(D^2 \mid \overline{\mathbf{P}}_i\right) = h\left(\left\| \overline{\mathbf{P}}_i - \overline{\mathbf{P}}_{avg} \right\|_2^2 \mid \overline{\mathbf{P}}_i\right), \tag{7}$$

when $\overline{\mathbf{P}}_{avg}$ contains a term of $\overline{\mathbf{P}}_i$, by isolating it, we have:

$$
\begin{aligned}
h\left(D^2 \mid \overline{\mathbf{P}}_i\right) &= h\left(\left\| \frac{n-1}{n}\overline{\mathbf{P}}_i - \frac{1}{n}\sum_{j=1,j\neq i}^n \overline{\mathbf{P}}_j \right\|_2^2 \mid \overline{\mathbf{P}}_i\right) \\
&= h\left(\left\| \overline{\mathbf{P}}_i - \frac{1}{n-1}\sum_{j=1,j\neq i}^n \overline{\mathbf{P}}_j \right\|_2^2 \mid \overline{\mathbf{P}}_i\right) + 2\log\left(\frac{n-1}{n}\right).
\end{aligned}
\tag{8}
$$

Notice $\overline{\mathbf{P}}_i$ and the normally distributed average, $\frac{1}{n-1}\sum_{j=1,j\neq i}^n \overline{\mathbf{P}}_j$, are independent. By taking the reduced prompt $\overline{\mathbf{P}}_i = \mathbf{p}$ as the conditional term, the first term of equation (8) is a non-central $\chi^2$-distribution with $d$ degrees of freedom and the non-centrality parameter $\tau$ is related to $\overline{\mathbf{P}}_i$ such that:

$$\tau \triangleq \sum_{k=1}^d \nu_k^2. \tag{9}$$

We denote each entry of $\frac{1}{n-1}\sum_{j=1,j\neq i}^n \overline{\mathbf{P}}_j$ as $X_k$, such that the random variable $D^2$ is:

$$D^2 \triangleq \sum_{k=1}^d \left(X_k + \nu_k\right)^2. \tag{10}$$

Then, $D^2$ is a non-central $\chi^2$-distribution with $d$ degrees of freedom and non-centrality parameter $\tau$. Apply the Theorem 1 in Moser (2020) using two families of function $g_m(\cdot), h_n(\cdot)$. We can reach the close form expression by:

$$
\begin{aligned}
h\left(\left\| \frac{1}{n-1}\sum_{j=1,j\neq i}^n \overline{\mathbf{P}}_j - \mathbf{p} \right\|_2^2 \mid \overline{\mathbf{P}}_i = \mathbf{p}\right) &= f_{\mathbf{P}}\left(\overline{\mathbf{P}}_i = \mathbf{p}\right) \mathbb{E}[-\log f(D^2)] \\
&= -f_{\mathbf{P}}\left(\overline{\mathbf{P}}_i = \mathbf{p}\right) \cdot \begin{cases} \ln(2) + h_d\left(\frac{\tau}{2}\right) + c & \text{if } d \in \mathbb{N}^{\text{odd}}, \\ \ln(2) + g_{d/2}\left(\frac{\tau}{2}\right) + c & \text{if } d \in \mathbb{N}^{\text{even}}. \end{cases}
\end{aligned}
\tag{11}
$$

Hence, the overall conditional entropy is:

$$h\left(\left\|\frac{1}{n-1}\sum_{j=1,j\neq i}^{n}\overline{\mathbf{P}}_j - \overline{\mathbf{P}}_i\right\|_2^2 \mid \overline{\mathbf{P}}_i\right) = \int\cdots\int_{\mathbf{p}\in\mathcal{P}} -f_{\overline{\mathbf{P}}_i}\left(\overline{\mathbf{P}}_i = \mathbf{p}\right)$$

$$\cdot \begin{cases} \left(\ln(2) + h_d\left(\frac{\tau}{2}\right) + c\right)\cdot d\mathbf{P} & \text{if } d\in\mathbb{N}^{\text{odd}},\\ \left(\ln(2) + g_{d/2}\left(\frac{\tau}{2}\right) + c\right)\cdot d\mathbf{P} & \text{if } d\in\mathbb{N}^{\text{even}}. \end{cases} \quad (12)$$

$f_{\overline{\mathbf{P}}_i}(\cdot)$ is the probability density function of $\overline{\mathbf{P}}_i$, i.e. a normal distribution; constant $c$ is $2\log\left(\frac{n-1}{n}\right)$. Combining two terms of $h\left(D^2\right)$ and $h\left(D^2\mid\overline{\mathbf{P}}_i\right)$, we have equation (3). Q.E.D.

(Part II)

In addition to Part I of the proof, we continue to discuss the other case when the prompt is outside of the given cluster. We continue from equation (5). When prompt $\overline{\mathbf{P}}_i$ is outside of the cluster, $h(D^2)$ is non-central $\chi^2$-distribution with $d$ degrees of freedom. The non-centrality parameter $\tau$ is defined by $\mu_i$. Hence, similar to equation (9), $\tau_i = d\cdot\mu_i^2$. Then:

$$h(D^2) = \begin{cases} \ln(2) + h_d\left(\frac{\tau_i}{2}\right) & \text{if } d\in\mathbb{N}^{\text{odd}},\\ \ln(2) + g_{d/2}\left(\frac{\tau_i}{2}\right) & \text{if } d\in\mathbb{N}^{\text{even}}. \end{cases} \quad (13)$$

For the other term of $h\left(D^2\mid\overline{\mathbf{P}}_i\right)$, different from equation (8), we do not need to isolate $\overline{\mathbf{P}}_i$. Therefore, there is no constant term $c$ in the final expression of the conditional entropy. Combining equation (13) and equation (12) without $c$ yields the counterpart of the theorem. Note that if the prompt distribution has $\mu_i = 0$, equation (13) still reduces to the first part of the proof.

$\square$

**Remark**: $g_m(\cdot)$ and $h_n(\cdot)$ are two family of functions introduced in Moser (2020). $g_m(\cdot):\mathbb{R}^+\to\mathbb{R}$ with $m\in\mathbb{N}$, which has the following expression:

$$g_m(\xi) \triangleq \begin{cases} \ln(\xi) - \text{Ei}(-\xi) + \sum_{j=1}^{m-1}(-1)'\left[e^{-\xi}(j-1)! - \frac{(m-1)!}{j(m-1-j)!}\right]\left(\frac{1}{\xi}\right)' & \text{if } \xi > 0,\\ \psi(m) & \text{if } \xi = 0. \end{cases} \quad (14)$$

where Ei is the exponential integral and $\psi$ is digamma function. $h_n(\cdot):\mathbb{R}^+\to\mathbb{R}$ with $n\in\mathbb{N}^{odd}$, which has the following expression:

$$h_n(\xi) \triangleq \begin{cases} -\gamma - 2\ln(2) + 2\xi\cdot {}_2F_2\left(1,1;\frac{3}{2},2;-\xi\right)\\ \quad + \sum_{j=1}^{\frac{n-1}{2}}(-1)^{j-1}\Gamma\left(j-\frac{1}{2}\right)\cdot\left[\sqrt{\xi}e^{-\xi}\,\text{erfi}(\sqrt{\xi}) + \sum_{i=1}^{j-1}\frac{(-1)^i\xi^i}{\Gamma\left(i+\frac{1}{2}\right)}\right]\left(\frac{1}{\xi}\right)^j & \text{if } \xi > 0,\\ \psi\left(\frac{n}{2}\right) & \text{if } \xi = 0. \end{cases} \quad (15)$$

where ${}_2F_2$ is a generalized hypergeometric function and erfi is the imaginary error function. For the analysis of the function families, please refer to Moser (2020).

### A.4 COMPLEXITY EVALUATIONS

We denote the LCC parameters by $\ell$ and the privacy threshold by $t = \alpha\cdot n$, where $\alpha$ is a constant allowing privacy against $\lfloor\alpha\cdot n\rfloor$ colluding users. The complexity is summarized in Table 3.

Table 3: Communication/Computation Overheads of SecFPP

| Communication | Computation | |
|---|---|---|
| User | User | Server |
| $\mathcal{O}(\frac{nd}{\ell} + kn)$ | $\mathcal{O}(\frac{d}{\ell}n\log^2 n + \frac{ndk+nd}{\ell})$ | $\mathcal{O}(kn^2\log^2 n)$ |

**User communication.** The communication overhead of each user consists of two parts: 1) the communication cost of secure secret sharing is $\mathcal{O}(\frac{nd}{\ell})$; 2) each client sends $\mathcal{O}(kn)$ distance shares to the server. The total communication cost of each client is $\mathcal{O}(\frac{nd}{\ell} + kn)$, which is dominated by $nd$. Here, $k$ is the number of cluster centers.

**User computation.** The computation performed by each user consists of three parts: 1) utilizing fast polynomial interpolation and evaluation Kedlaya & Umans (2011), each user generates the secret shares of its local prompt with complexity $\mathcal{O}(\frac{d}{\ell}n\log^2 n)$; 2) Within each round, each user first updates the secret shares of $k$ centers, and computes the distances from each data point to each center in the finite domain, taking $\mathcal{O}(\frac{ndk}{\ell})$ operations. 3) Secure aggregation by $\mathcal{O}(\frac{nd}{\ell})$ The total computation complexity of each client is $\mathcal{O}(\frac{d}{\ell}n\log^2 n + \frac{ndk+nd}{\ell})$, which is dominated by $nd\log^2 n$.

**Server computation.** Server's computation complexity consists of two parts: 1) The decoding of the actual pair-wise distances takes $\mathcal{O}(kn^2\log^2 n)$ operations; 2) The cost of running cluster assignment on each prompt is $\mathcal{O}(kn)$. Thus, the total computation complexity of the server over $s$ iterations is $\mathcal{O}(kn^2\log^2 n)$.

Next, we present the empirical overheads of performing SecPC in Figure 2 with dominant factors: $n$ and $d$. Privacy threshold $\alpha$ is set to $1/3$ consistently and $\ell = \lfloor \frac{n-t}{2} \rfloor$ accordingly. The large-enough prime $q$ is sampled in the scale of $10^{10}$ and the quantization parameter $\lambda$ is $10^3$ (see Appendix A.1 for quantization details). As shown in Figure 2, the computational cost is almost linearly predominated by $nd$ and the user's overhead is the primary cost. Though the server computation complexity has $n^2\log^2 n$ term, its overhead is negligible in practice. The user's computation time reaches $0.4$s when the dimension is $4000$, however, when executing SecFPP, SecPC uses reduced prompts with rank less than 10, multiplying the number of tokens, the total dimension is only $d = 150$. For communication costs, consider a 4G network with 98 Mbps bandwidth and the largest parameter settings in experiments. It takes $0.05$s for a user to share and less than $10^{-4}$s to communicate distances, which is even faster with networks like LAN or 5G. Overall, the communication and computation overheads additionally for privacy preservation are negligible.

## A.5 REPRODUCIBILITY STATEMENT

The source code is provided in the supplementary material. To ensure reproducibility, please follow the instructions in the *README.md* file to set up the environment using *requirements.txt*, and to download the necessary pre-trained models and datasets. All experiments can then be reproduced by executing the provided *run.sh* script with the specified settings as mentioned in Appendix A.2.

## A.6 THE USE OF LLMS

We disclose the strictly limited utilization of GenAI tools exclusively for linguistic parts of originally human-authored content. No GenAI is employed in the research ideation or creation or modification of figures, tables, flowcharts, or any non-textual elements. The implementation adhered to the following guidelines: authors use DeepSeek and ChatGPT with Prompts like

- Use academic writing to refine the following text:
- Proofread the following text:
- Polish the following text:

All AI-processed contents are sifted by human verification to ensure academic authenticity and factual accuracy. The authors are fully accountable for all parts of the paper.

