# OpenReview forum: "SecFPP : Secure Federated Prompt Personalization for Vision Language Models"
_ICLR.cc/2026/Conference — ICLR 2026 Conference Withdrawn Submission_

### Official Review · Reviewer_q7fU · 2025-10-23

**Soundness:** 1
**Presentation:** 1
**Contribution:** 2
**Rating:** 2
**Confidence:** 5

**Summary:**

The paper proposes SecFPP, a federated prompt personalization framework that provide formal privacy. SecFPP operates as a distributed k-mean clustering algorithm and provides privacy via secure aggregation protocol. In SecFPP framework, each user maintains a local prompt and a shared federated prompt that is updated via k-mean clustering. In each federated communication round, each user secret shares their truncated personalized prompt and federated prompt using Lagrange coded computation. The server maintains and updates a cluster assignment which is distributed to all users every round.

**Strengths:**

* Motivation is well stated, and the problem the paper tries to address is important in multimodal LLM framework.
* Experiment setting is clear and described in detail. The performance results support the author's claims.

**Weaknesses:**

* While the idea is interesting, the proposed method is just a combination of federated k-mean clustering with secure aggregation, both of which have been extensively studied in the past decades, limiting the novelty.

* Algorithm 1 is vague with a lot of missing details. What are $\nabla_{G,i} \mathcal{L}$ and $\nabla_{L,i} \mathcal{L}$ and how are they computed? They are not introduced anywhere in the paper. Step 7 is very confusing and incomplete. What exactly does $PCA(.)$ compute? How is $P_i^{(t+1)}$ computed? It seems that $P_i^{(t+1)}$ is not known until step 12.

* The author points out the existing challenge of gradient-based privacy attacks in federated learning setting (line 150), however, the proposed method does not prevent such attacks. The shared components, i.e. personalized prompt and federated gradients, may have information about the gradient of the local components. This is a serious gradient leakage which is susceptible to gradient-based attacks.

* I disagree with the author's claim that the proposed method has negligible overheads (line 90). There are lots of communication overheads in SecFPP compared to a standard federated learning framework and other FPP baselines. First, users share both the personalized prompt $\bar{P_i}$ (Algorithm 1 step 9) and the federated gradients (Algorithm 1 step 10) to the server, while previous FPP baselines only need to share the latter. Second, the secure aggregation protocol requires user-to-user in addition to user-to-server communication as shown in Algorithm 2, while other FPP methods only require user-to-server communication. Furthermore, the protocol is applied twice to the personalized prompt and the federated gradients, adding extra overheads.

* While dimension reduction may reduce some communication overhead, per-round SVD computation is expensive. A detailed communication-computation cost tradeoff should be discussed to justify if the use of truncated SVD brings any benefit.

**Questions:**

* What does $F(.)$ compute mathematically? How is the loss computed when $P_i$ involves two separate components?

* What are the dimensions of $P_i$ before and after dimension reduction?

* How is the accuracy in Table 2 computed? How can one interpret the personalization and generalization based on the results, especially for single datasets (class heterogeneity)?

* What are the communication and computation overheads of the proposed method compared to other baselines? It would be helpful if the author can show the overhead comparison theoretically and empirically.

* Does SecFPP provide equivalent privacy guarantee to DP-FPL? How does one quantify the privacy provided by secure aggregation and differential privacy?

---

### Official Review · Reviewer_ZHqX · 2025-10-31

**Soundness:** 3
**Presentation:** 3
**Contribution:** 3
**Rating:** 6
**Confidence:** 3

**Summary:**

The paper proposes SecFPP, a framework for secure federated prompt personalization that combines secret sharing and Lagrange coded computation. Prompts are decomposed into federated and local parts, and a privacy-preserving clustering method (SecPC) aggregates prompts within domains. Mutual information analysis supports the theoretical privacy claims, and experiments under multi-domain and non-IID settings show strong performance even under strict privacy constraints.

**Strengths:**

1. The work addresses a timely and practically important problem by integrating privacy-preserving computation with prompt personalization.

2. The proposed design is modular and interpretable, with each component clearly targeting a specific type of heterogeneity.

3. Theoretical analysis and empirical results complement each other, providing a balanced and convincing evaluation.

**Weaknesses:**

1. The security analysis relies on the honest-but-curious assumption, leaving stronger adversarial scenarios such as collusion or data manipulation insufficiently discussed.

2. The protection mechanism in the clustering stage is not fully clear;  small or imbalanced clusters could expose additional information, which warrants further analysis.

3. Quantization and finite-field operations may introduce nontrivial numerical errors, yet their effects on model performance and security are not systematically evaluated.

4. A more comprehensive exploration of hyperparameters would be valuable, including systematic experiments and analysis of how different settings affect overall performance.

5. The scalability discussion remains brief;  the communication and computation overhead of large-scale deployments should be further examined.

**Questions:**

1. How does SecFPP handle adversarial settings involving collusion or active attacks, and what is the tolerable number of compromised participants?

2. How are very small or imbalanced clusters treated in SecPC, and could they cause additional leakage risks?

3. What guidelines exist for choosing the quantization parameters (λ, q), and how sensitive is the performance to these settings?

---

### Official Review · Reviewer_PpxW · 2025-10-31

**Soundness:** 1
**Presentation:** 3
**Contribution:** 2
**Rating:** 2
**Confidence:** 4

**Summary:**

This paper presents a new federated algorithm for prompt personalization for vision-language models. The algorithm, SecFPP, uses a novel secure federated clustering method to group clients, and each client refines its global prompt within its own cluster. The experimental results show, in general, higher accuracy than a number of non-private baselines and one private baseline.

**Strengths:**

- The paper presentation is overall clear and easy to follow.
- The SecPC method and its analysis are novel and may have broader applications.
- SecFPP demonstrates impressive accuracy in experiments.

**Weaknesses:**

- The paper compares SecFPP to a differentially private method DP-FPL as a privacy baseline, however, SecFPP does not provide the same privacy guarantees. The DP method provides quantifiable privacy guarantees for the training data over the entire training process. While SecFPP gives a statistical measure of the privacy of the training data in SecPC, this is not as strong a guarantee as DP. More importantly, SecFPP leaks information about the training data though the aggregated gradient of the global prompts in line 10 of Algorithm 1. The secure aggregation does protect the inputs to the aggregate, but the aggregated value is still a privacy leak. This is a major issue as the method fails at its main claim of privacy.
- Even if SecFPP does not protect privacy, the experiment results indicate it may still be of value for personalized prompt learning. The results do not really give a good indication of the balance between personalization and generalization. It would be helpful to break up the results in the multi-domain setting to differentiate performance on own domain vs. other domain.
- Minor: there appears to be something wrong with the citations that are not used inline in sentences.
- Minor: For Table 1, described in Section 2.2, please include the dataset details.

**Questions:**

- What is the intuition behind the benefit of the clustering. Is there any way to demonstrate this through the experiments?

---

### Note · Authors · 2025-11-26

I have read and agree with the venue's withdrawal policy on behalf of myself and my co-authors.